# First insights into coral recruit and juvenile abundances at remote Aldabra Atoll, Seychelles

Anna Koester[1,2]*, Amanda K. Ford[3], Sebastian C. A. Ferse[1,4], Valentina Migani[5], Nancy Bunbury[2,6], Cheryl Sanchez[2,7], Christian Wild[1]

1 Marine Ecology Department, Faculty of Biology & Chemistry, University of Bremen, Bremen, Germany, 2 Seychelles Islands Foundation, Victoria, Mahé, Seychelles, 3 School of Agriculture, Geography, Environment, Ocean and Natural Sciences, University of the South Pacific, Suva, Fiji, 4 Leibniz Centre for Tropical Marine Research, Bremen, Germany, 5 Institute for Ecology, Faculty of Biology & Chemistry, University of Bremen, Bremen, Germany, 6 Centre for Ecology and Conservation, University of Exeter, Cornwall Campus, Penryn, United Kingdom, 7 Department of Biology, University of Pisa, Pisa, Italy

* a.koester@mail.de

**Data Availability Statement:** All relevant data are within the manuscript and its Supporting information files.

## Abstract

Coral recruitment and successive growth are essential for post-disturbance reef recovery. As coral recruit and juvenile abundances vary across locations and under different environmental regimes, their assessment at remote, undisturbed reefs improves our understanding of early life stage dynamics of corals. Here, we first explored changes in coral juvenile abundance across three locations (lagoon, seaward west and east) at remote Aldabra Atoll (Seychelles) between 2015 and 2019, which spanned the 2015/16 global coral bleaching event. Secondly, we measured variation in coral recruit abundance on settlement tiles from two sites (lagoon, seaward reef) during August 2018–August 2019. Juvenile abundance decreased from 14.1 ± 1.2 to 7.4 ± 0.5 colonies m$^{-2}$ (mean ± SE) during 2015–2016 and increased to 22.4 ± 1.2 colonies m$^{-2}$ during 2016–2019. Whilst juvenile abundance increased two- to three-fold at the lagoonal and seaward western sites during 2016–2018 (from 7.7–8.3 to 17.3–24.7 colonies m$^{-2}$), increases at the seaward eastern sites occurred later (2018–2019; from 5.8–6.9 to 16.6–24.1 colonies m$^{-2}$). The composition of coral recruits on settlement tiles was dominated by Pocilloporidae (64–92% of all recruits), and recruit abundance was 7- to 47-fold higher inside than outside the lagoon. Recruit abundance was highest in October–December 2018 (2164 ± 453 recruits m$^{-2}$) and lowest in June–August 2019 (240 ± 98 recruits m$^{-2}$). As Acroporid recruit abundance corresponded to this trend, the results suggest that broadcast spawning occurred during October–December, when water temperature increased from 26 to 29°C. This study provides the first published record on coral recruit abundance in the Seychelles Outer Islands, indicates a rapid (2–3 years) increase of juvenile corals following a bleaching event, and provides crucial baseline data for future research on reef resilience and connectivity within the region.

**Funding:** The authors received no specific funding for this work.

**Competing interests:** The authors have declared that no competing interests exist.

## Introduction

Coral recruitment and juvenile abundances are important indicators of reef resilience [1, 2]. Large-scale disturbances such as mass coral bleaching events have driven major losses of live coral worldwide, causing substantial changes to coral reef ecosystem structure and functioning [3, 4]. Coral reefs can recover from such disturbances, and the reassembly of scleractinian coral communities, especially habitat-forming species, is particularly important for the recovery of ecosystem functions [2, 5, 6]. Reef recovery depends on the survival and growth of remnant coral colonies and on coral recruitment, and understanding the dynamics of early life stages can provide valuable insights into potential post-disturbance shifts in coral community composition [5, 7–9].

Scleractinian corals, like most sessile marine invertebrates, have a bipartite life history. After a free-swimming pelagic larval phase and successful larvae settlement, the coral develops from recruit (< 1 cm diameter) to juvenile (< 5 cm diameter) to adult during its sessile benthic phase [10, 11], a process referred to as 'recruitment'. A prerequisite for recruitment is the sufficient supply of coral larvae, originating from within (self-recruitment) or outside the local coral community. Self-recruitment (see [12]) is particularly important for geographically isolated reefs [6] and appears to be more common among brooding corals [8, 13]. Well-connected reefs benefit from the influx of external coral larvae, especially if the local adult coral population has been severely degraded [14]. In addition to larval supply, the availability of suitable benthic substrate for settlement and post-settlement survival plays a key role in coral recruitment [15]. For example, while crustose coralline algae (CCA) can promote recruitment by inducing larval settlement and by facilitating survival and growth of settlers [16, 17], algal turfs (if not cropped short [18]) and macroalgae can inhibit coral larval settlement and are aggressive competitors for space [19]. Unstable substrates (e.g. unconsolidated rubble) are also unsuitable, as their movement on the seafloor can cause major coral recruit and juvenile die-offs [20]. Post-settlement mortality of corals is generally high until they have reached sizes of > 5 cm [8], when they are more likely to withstand predation by corallivores or incidentally by herbivores [21, 22], and competition with other benthic organisms (e.g., turf and macroalgae [16, 23]).

Anthropogenic stressors, such as nutrient enrichment, coastal development, and overfishing, hamper coral recruitment through various processes, for example by altering the transmission of chemical signals involved in coral reproduction [24], reducing fertilisation of coral larvae due to high sedimentation [25, 26], and facilitating the proliferation of algal turfs and macroalgae [27–29]. However, even on relatively undisturbed reefs, mortality and varying coral growth rates influence the community structure of juvenile and adult corals and the trajectories of coral recovery following major disturbances [11, 30, 31].

Aldabra Atoll, in the Western Indian Ocean (WIO), provides an opportunity to study coral recruitment and coral reef recovery under conditions of minimal human disturbance. Aldabra was designated as a Special Reserve in 1981 (the highest level of national protection under Seychelles' legislation) and was inscribed as a UNESCO World Heritage Site in 1982. Its marine ecosystem has therefore been protected from commercial fishing pressure for almost 40 years, there is no coastal development, and human-driven nutrient inputs are absent. Nevertheless, coral bleaching events have caused high coral mortality at Aldabra in 1998/1999 (38–66% mortality at 10–20 m water depth [32]) and in 2015/2016 (34–62% mortality at 2–15 m water depth [33, 34]). Furthermore, while the reefs at 2–5 m water depth recovered 54–93% of their pre-bleaching (2015/2016) coral cover by 2019, no recovery was observed at 15 m water depth [34].

To better understand temporal and spatial patterns of corals in early life stages under minimal anthropogenic disturbance, we measured changes in abundance of coral juveniles across

three reef locations at Aldabra Atoll between 2015–2019, covering different reef depths and habitats. This not only provided us with multiple years of data on juvenile coral dynamics, but also spanned a mass-bleaching event (2015/2016) and thus provided early insights into post-disturbance reef recovery at this remote location. We also periodically analysed settlement tiles during 2018–2019 to document, for the first time, patterns of coral recruit abundance within the Outer Islands of the Seychelles. Our work provides valuable baseline data for future research on the recovery of these reefs and reef connectivity within Aldabra and the broader region.

## Materials and methods

### Study site

Aldabra (46˚20'E, 9˚24'S) is an elevated coral atoll in the south-west of the Seychelles archipel-ago (Fig 1). Aldabra spans 34 × 14.5 km and consists of four main islands that encircle a 203

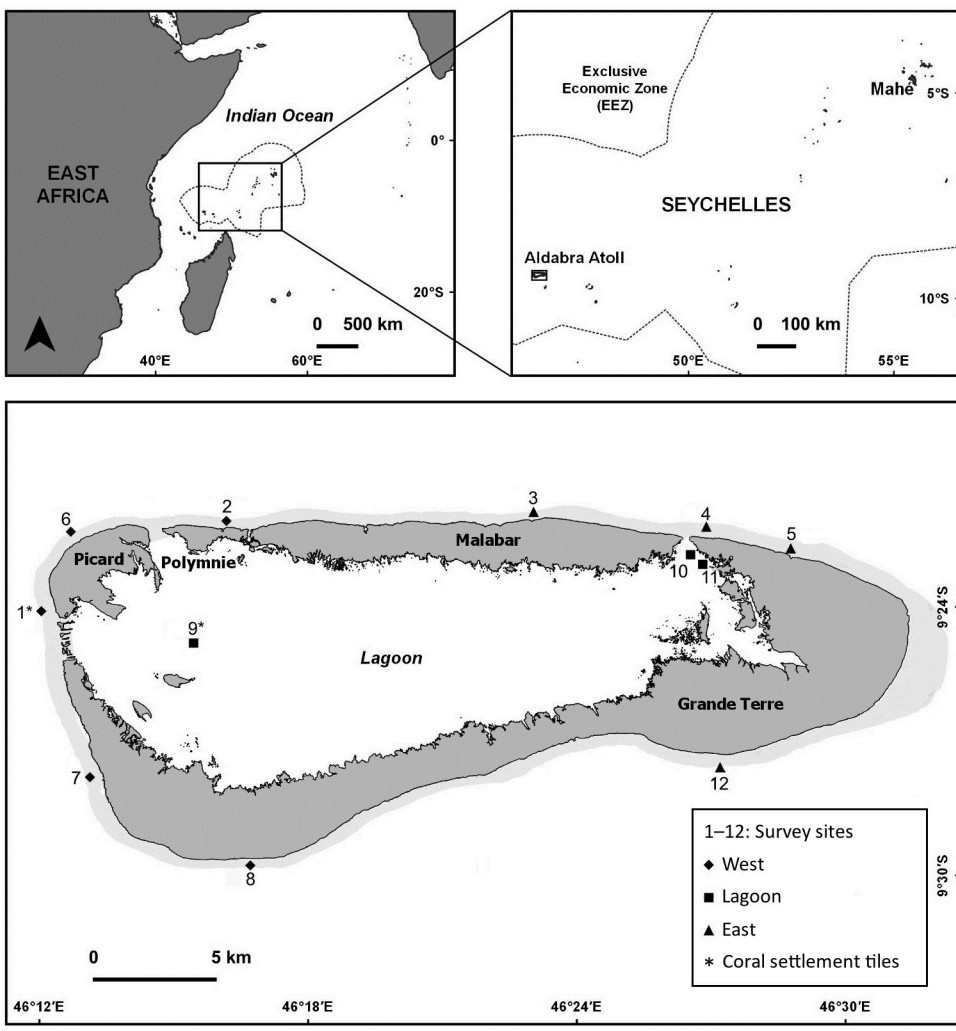

**Fig 1. Location of Aldabra Atoll.** Location of Aldabra Atoll in the Indian Ocean and within the Seychelles (top) with its four main islands and 12 survey sites (represented by numbers) at the seaward western ('West', n = 5), seaward eastern ('East', n = 4) and lagoonal ('Lagoon', n = 3) reefs (bottom). Figure included under a CC BY license, with permission from the Seychelles Islands Foundation, copyright 2021.

km$^2$ large lagoon [35]. During April–November, Aldabra is usually characterised by a drier and cooler climate, whilst the north-west monsoon during November/December–March creates wetter and warmer conditions.

There are 12 permanently marked marine survey sites around Aldabra [33]. Of these, nine sites are located on the seaward reefs (five sites at seaward west—Sites 1, 2, 6, 7, 8; four at seaward east—Sites 3, 4, 5, 12; sites surveyed at 5 and 15 m water depth) and three sites are located at *ca*. 2 m water depth in the lagoon (Sites 9–11) (Fig 1). For the purposes of this study, the 2 m and 5 m depths are referred to as 'shallow' and 15 m as 'deep'. The division of the seaward reef sites into 'west' and 'east' is based on the different wind and wave exposure on either side of Aldabra and the differing benthic community compositions [34]. Although both locations are subject to strong currents, the north-west and west of Aldabra are relatively sheltered throughout most of the year, while the north-east, east and south-east are more exposed to strong winds and high wave energy [36].

Between December 2015 and June 2016, Aldabra's sea surface temperatures reached a maximum of 30.7˚C (March 2016), and the reefs experienced continuous bleaching risk as indicated by the degree heating week values which peaked at 3.4˚C-weeks (satellite derived temperature measured *ca*. 55 km north-east of Aldabra: 46˚50'E, 9˚00'S [33]). By December 2016, hard coral cover had been reduced by 34% at 2 m water depth in the lagoon (from 32% cover in 2014 to 21% in 2016) and by 51–62% on the seaward reefs at 5 m and 15 m water depth (from 10–34% cover in 2014 to 5–15% in 2016 [33, 34]). By 2019, hard coral cover had not changed significantly at Aldabra's deep locations, which remained dominated by turf algae and the calcifying macroalgae *Halimeda*. In the shallows, where turf algae had reduced to pre-bleaching levels, coral cover had increased significantly and reached 54% (seaward east, 5 m depth), 68% (seaward west, 5 m depth) and 93% (lagoon) of the pre-bleaching hard coral cover [34].

Approval for this research was granted by the Seychelles Bureau of Standards (Reference: A0157).

## Data collection

**Coral juveniles.** At each of the 12 survey sites, juvenile coral abundance (i.e., colonies < 5 cm diameter) was assessed along three 20-m transects separated by a 10-m gap (except in 2015, when only one or two transects were conducted at each deep site; S1 Table). Small fragmented adult colonies were distinguished from juvenile corals by checking for the original colony in the proximity and observing the colony form. Five 0.25-m$^2$ quadrats were randomly placed along each transect (stratified, to avoid duplication), within which the number of all coral juveniles was recorded and identified to the lowest possible level, i.e., genus, or family where further identification was not possible. Surveys were conducted in December 2015 (immediately before the bleaching event), December 2016 (six months after the end of the bleaching event), December 2018–January 2019 (hereafter 2018) and November 2019–January 2020 (hereafter 2019). This resulted in a total of 1085 quadrats being assessed. Limited resources meant that survey effort varied across years and no 2015 data is available for the lagoonal reefs (replication detailed in S1 Table).

**Coral recruits.** The abundance of coral recruits (i.e., everything visible and identifiable as coral) was assessed on the unglazed side of ceramic tiles (15 × 15 × 1 cm). Tiles were attached to half concrete masonry blocks with stainless steel bolts and spacers (following [20]), with the unglazed side facing down. As coral recruits, particularly at shallow depths, settle predominantly on the sheltered side of settlement tiles [15, 37], assessments were restricted to the unglazed, underside of the tile. Between August 2018 and August 2019, tiles were placed at one lagoonal (Site 9) and one western seaward site (Site 1; 5 m depth; Fig 1).

These sites were selected because they are accessible for diving throughout the year. Tiles were replaced every two months and retrieved tiles were submerged in household bleach (10% solution) for 24 h, rinsed in freshwater, and dried before the unglazed underside was inspected under a stereo microscope. Coral recruits were counted and classified into three taxonomic categories (Acroporidae, Pocilloporidae, or 'other'; following [38]). Between nine and 14 tiles were assessed per survey period for each site, as occasional storms damaged individual tiles at Site 1 (S2 Table).

**Water temperature.** Water temperature data (recorded every 30 min) was obtained from two Onset loggers (HOBO U22-001, accuracy: ± 0.2˚C [39]) deployed at Site 9 (lagoon) between 10 February 2015 and 14 November 2018, and at Site 1 (seaward west, 5 m) between 16 December 2013 and 31 December 2019. Due to technical issues, there was a 3-week data gap at Site 9 (10 December 2016–1 January 2017), and two 1-week data gaps (2–9 April 2015; 23–30 December 2018) and one 5-month data gap (12 December 2016–5 May 2017) at Site 1.

## Statistical analysis

Statistical analysis was conducted using R version 3.6.1 [40]. All plots were created with the *ggplot2* package [41].

**Coral juveniles.** We pooled coral juveniles into the families Acroporidae, Pocilloporidae, Poritidae, Merulinidae, and Agariciidae following the World Register of Marine Species [42]. *Leptastrea* was included as an individual genus due to its high abundance at Aldabra (9% of overall counts) and because it is not assigned to any family as it is considered as *Incertae sedis* at the family level [43]. Combined, these taxa comprised 74% of overall counts; all other taxa were grouped into 'other'.

We used Generalised Linear Mixed Models (GLMM, *lme4* package [44]) with Poisson or negative binomial error distributions to test for differences in coral juvenile abundance across years (2015, 2016, 2018, 2019), locations (lagoon, east, west) and water depths (shallow, deep; S3 Table). GLMMs are designed to handle unbalanced study designs and can therefore account for the missing lagoonal data in 2015 and the variable number of survey sites per location. Models with overall coral juvenile abundance and abundances of Acroporidae, Pocilloporidae, Merulinidae, Agariciidae and 'other' as response variables included 'survey year', 'location', 'depth' and their interaction as fixed factors. Because models with Poritidae and *Leptastrea* as response variables had a poor fit when the interaction of fixed effects was included, only differences across years were tested, i.e. only 'survey year' was set as fixed factor. In all models, 'transect' nested in 'survey site' was set as a random factor to account for possible autocorrelation between quadrats. Since zero-inflation is a typical feature of count data, we checked for an excess of zeros in the coral juvenile dataset (none detected) and validated the model residuals with the *DHARMa* package. This package uses a simulation-based approach to create standardised residuals for generalised linear (mixed) models and provides additional test functions to detect typical model specification problems [45]. We conducted a post-hoc analysis based on least square means with Bonferroni adjustment (*lsmean* package [46]) to identify pairwise differences between the variables in significant models.

**Coral recruits.** We used a GLMM to test for differences in coral recruit abundance across survey period (August–October 2018, October–December 2018, December 2018–February 2019, February–April 2019, April–June 2019, June–August 2019) and location (Site 1: seaward site, Site 9: lagoonal site; S4 Table). Fixed factors included survey period, location, and their interaction with location set as a random factor. The initial Poisson model was overdispersed (but not zero-inflated), and the final model was specified with negative binomial error distribution using the *glmmTMB* package [47]. Model validation and post-hoc analysis was done in

the same way as described for the coral juvenile analysis (using the *DHARMa* and *lsmean* packages, respectively).

**Coral juvenile and recruit abundance in relation to coral coverage.** To evaluate patterns in coral juvenile and recruit abundance in relation to Aldabra's hard coral cover, percentage hard coral cover values were obtained from previous work conducted at Aldabra [34] with coral categories re-assigned to match those described here (i.e. Acroporidae and Pocilloporidae for recruits and juveniles, and Poritidae, Merulinidae, Agariciidae and *Leptastrea* for juveniles). Data was plotted (S1 Fig) and compiled in Tables 1 and 2 for visual comparison with the data on coral juvenile and recruit abundance.

**Aldabra's coral juvenile abundance in a global context.** We compiled a table of studies reporting coral juvenile abundances (S5 Table), including only research which assessed coral juveniles of < 5 cm diameter without further visual aid.

## Results

### Spatial and temporal trends in coral juvenile abundance and composition

The abundance of coral juveniles changed significantly throughout the study period, with an initial reduction from 2015 to 2016, and a successive increase by 2019 (Fig 2a). Between 2015 and 2016 (immediately before and after the mass bleaching event, respectively), overall coral juvenile abundance at Aldabra's seaward reefs decreased by 48%, from 14.1 ± 1.2 to 7.4 ± 0.5 colonies m$^{-2}$ (mean ± SE), but reductions differed across locations and water depths (Fig 2; S3 Table). At western sites, abundances dropped by 60% in shallow and 46% in deep water, although the decline was not significant at the latter (Fig 2b; Table 1). At the more exposed eastern sites in shallow water, abundances did not change significantly between 2015 and 2016, although notably they were low in 2015, matching abundances recorded at all other locations in 2016 (i.e., after the bleaching event). At the deep eastern sites, abundances significantly dropped by 52% (Fig 2c, Table 1), although these results should be treated with caution due to

**Table 1. Information on coral juvenile abundances and percentage hard coral cover at Aldabra between 2014 and 2019.** Asterisks indicate significant differences between years; n.a. = not available.

| Measure | Location | Water depth | Pre-bleaching[a] 2014/15 | | Post-bleaching 2016 | | Post-bleaching 2019 | | % Change 2014/15–2016 | % Change 2016–2019 | % Recovery[b] |
|---|---|---|---|---|---|---|---|---|---|---|---|
| | | | Mean | SE | Mean | SE | Mean | SE | | | |
| Coral juveniles (no. m$^{-2}$) | West | Shallow | 20.6 | 1.9 | 8.3 | 1.0 | 21.4 | 1.9 | -60 * | 61 * | 104 |
| | West | Deep | 14.3 | 5.1 | 7.7 | 0.8 | 22.9 | 1.8 | -46 * | 66 * | 160 |
| | East | Shallow | 8.3 | 0.8 | 7.9 | 1.0 | 24.1 | 2.6 | -6 * | 67 * | 290 |
| | East | Deep | 11.5 | 2.7 | 5.5 | 1.0 | 16.6 | 1.9 | -52 * | 67 * | 144 |
| | Lagoon | Shallow | n.a. | n.a. | 7.7 | 1.6 | 29.9 | 5.3 | n.a * | 74 * | n.a |
| Coral cover (%)[c] | West | Shallow | 34.1 | 2.9 | 15.9 | 1.8 | 23.7 | 2.0 | -53 * | 33 * | 69 |
| | West | Deep | 20.8 | 1.9 | 8.6 | 1.0 | 13.6 | 1.2 | -59 * | 37 * | 65 |
| | East | Shallow | 23.6 | 4.2 | 9.7 | 0.8 | 12.5 | 1.5 | -59 * | 22 * | 53 |
| | East | Deep | 10.1 | 2.9 | 5.1 | 1.1 | 5.0 | 1.2 | -49 * | -0.02 * | 55 |
| | Lagoon | Shallow | 32.3 | 5.0 | 19.6 | 4.5 | 30.0 | 4.7 | -39 * | 35 * | 93 |

[a] 2014: Coral cover; 2015: Coral juveniles;

[b] Percentage of pre-bleaching value reached in 2019;

[c] Significant changes based on previous work conducted at Aldabra [34]; note, displayed here are mean coral cover values and the respective percentage changes and not back-transformed model estimates and the respective percentage changes shown in previous study [34].

**Table 2. Percentage coral cover, coral juvenile abundances and coral recruit abundances at the two study sites in Aldabra's lagoon and the seaward west.**

| Location | Taxa | Coral cover (%) 2019 | | Coral juveniles (no. m⁻²) 2019 | | Coral recruits (no. m⁻²) Aug 2018–Aug 2019 | |
|---|---|---|---|---|---|---|---|
| | | Mean | SE | Mean | SE | Mean | SE |
| Lagoon (Site 9) | Acroporidae | 21.0 | 3.6 | 18.9 | 4.2 | 365.6 | 59.9 |
| | Pocilloporidae | 11.0 | 8.9 | 14.4 | 5.1 | 1545.7 | 138.9 |
| | Other | 10.4 | 3.2 | 34.7 | 8.3 | 171.1 | 21.1 |
| | Total | 42.4 | 4.8 | 68.0 | 9.8 | 2079.6 | 193.3 |
| Seaward west (Site 1) | Acroporidae | 7.2 | 3.5 | 2.1 | 0.9 | 64.1 | 15.6 |
| | Pocilloporidae | 0.6 | 0.2 | 1.9 | 0.7 | 44.4 | 7.5 |
| | Other | 6.6 | 1.9 | 13.3 | 2.9 | 32.4 | 15.3 |
| | Total | 14.4 | 5.4 | 17.3 | 3.2 | 141.0 | 26.4 |

low replication (S1 Table). From 2016 to 2019, coral juvenile abundance at all locations and all depths showed a positive trend, tripling over this timeframe from 7.4 ± 0.5 to 22.4 ± 1.2 colonies m⁻² and reaching or exceeding levels recorded in 2015 (i.e. pre-bleaching) at all seaward sites (Fig 2; Table 1; S3 Table). Whilst mean coral juvenile abundance increased two- to three-fold at the lagoon and western sites between 2016 and 2018, a similar increase occurred later—between 2018 and 2019—at the eastern sites (Fig 2).

Overall, abundance of juvenile Acroporidae, Merulinidae, Agariciidae and 'other' corals declined significantly between 2015 and 2016 (mean abundances dropping by 66%, 54%, 73% and 63%, respectively; Fig 3a–3g; S3 Table), with reductions differing among taxa and locations (Fig 3h–3u). Contrastingly, abundance of Pocilloporidae and *Lepastrea* corals remained unchanged, and Poritidae corals exhibited an increase during this time. At the western sites (Fig 3h–3n), declines in coral juvenile abundance were significant for Acroporidae (reduction of mean abundance: shallow: -60%, deep: -60%), Merulinidae (shallow: -45%), Agariciidae (shallow: -88%) and 'other' (shallow: -79%, deep: -71%). At the eastern sites, only the abundance of Merulinidae juveniles declined significantly between 2015 and 2016 (deep: -86%; Fig 3r). During 2016–2019, mean abundance of juveniles of all taxa (except Merulinidae) reached or exceeded 2015 levels at all seaward sites during 2016–2019 (Fig 3h–3u). Overall mean abundances were many times higher in 2019 than in 2015 for Poritidae (14-fold; Fig 3c) and *Leptastrea* (nine-fold; Fig 3f), while mean abundances of Merulinidae were lower in 2019 than 2015 at all seaward sites (Fig 3k and 3r). In the lagoon, abundances of Acroporidae and 'other' juveniles increased during 2016–2019 (Fig 3v and 3B), with mean abundances of Acroporidae,

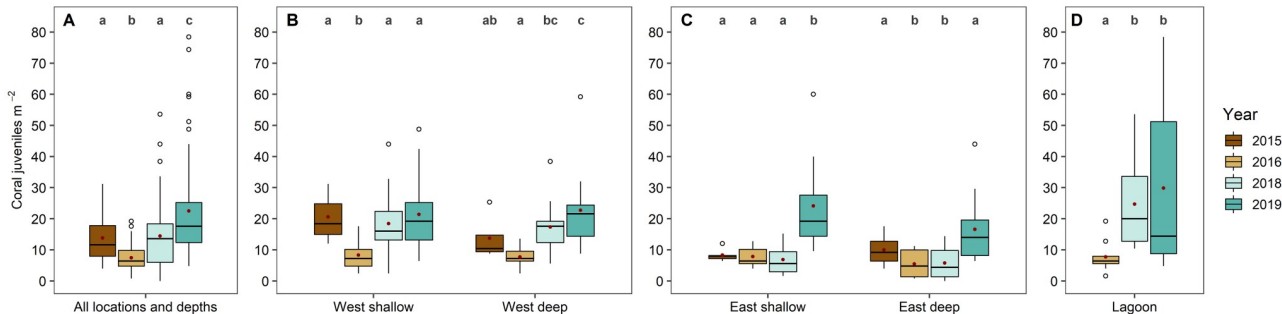

**Fig 2. Coral juvenile abundance across locations and years.** A: Across all locations and depths; B: Seaward west (2015–2019); C: Seaward east (2015–2019); D: Lagoon (2016–2019). Plots show median and interquartile range with outliers displayed as circles and means displayed as red dots. Small letters above boxplots indicate significant differences ($p < 0.05$) in abundances between years within locations and depths (see S3 Table).

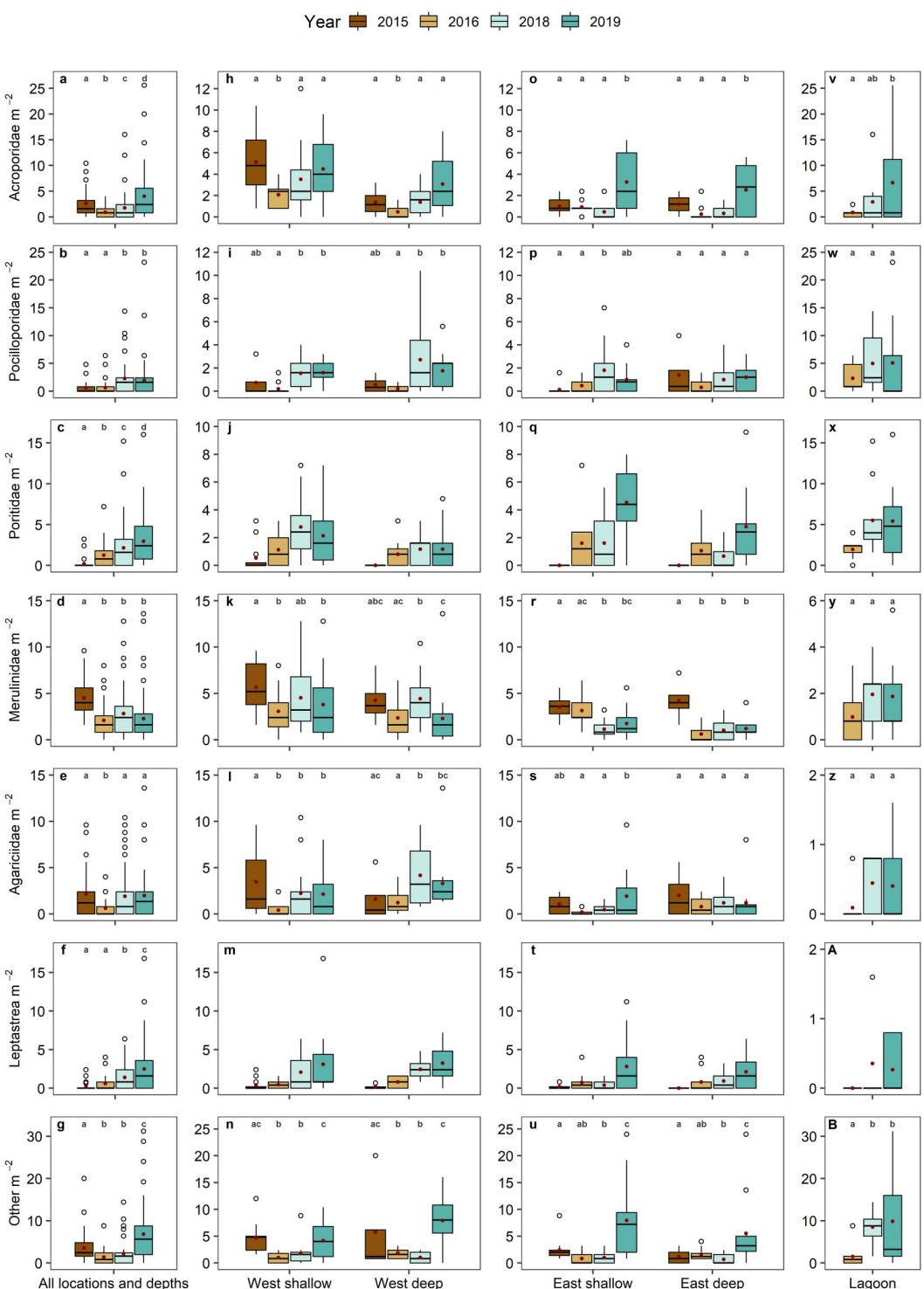

**Fig 3. Coral juvenile abundance of the most abundant taxa across locations and years.** Displayed are coral juvenile abundances across all locations and depths (a–g) and at the seaward locations (h–u) during 2015–2019 and the lagoon (v–B) during 2016–2019. Plots show median and interquartile range with outliers displayed as circles and means displayed as red dots. Note the different y-axis scaling. Small letters above boxplots indicate significant differences (*p* < 0.05) in abundances between years within locations and depths (see S3 Table). Note that only overall changes across years could be statistically tested for Poritidae and *Leptastrea* (see Methods).

Pocilloporidae, and Poritidae being relatively similar in 2019 (5.0–6.5 colonies m$^{-2}$; Fig 3v–3x). Abundances of 'other' juveniles were highest in 2019 (mean: 10.1 colonies m$^{-2}$; Fig 3B), while few Merulinidae, Agariciidae, and *Leptastrea* juveniles were recorded in the lagoon in all years (mean: 0.3–2.0 colonies m$^{-2}$; Fig 3y–3A).

### Spatial and temporal trends in coral recruit abundance and composition

Between August 2018 and August 2019, a total of 3591 coral recruits were counted on 142 tiles. Pocilloporidae comprised 72% of all counts, followed by Acroporidae (19%) and 'other' (9%). Of all recruits counted, 93% occurred on the tiles at the lagoonal site (3369 recruits on 72 tiles; 2080 ± 193 recruits m$^{-2}$; mean ± SE; Table 2), whilst 7% were recorded on the tiles at the seaward site (222 recruits on 70 tiles; 141 ± 26 recruits m$^{-2}$).

Overall coral recruit abundance ranged from 240 ± 98 recruits m$^{-2}$ (mean ± SE) in June–August 2019 to 2164 ± 453 recruits m$^{-2}$ in October–December 2018 (Fig 4a). Recruit abundance was seven- to 47-fold higher at the lagoonal than the seaward site throughout the study (Fig 4a), with differences being significant for all but the June–August 2019 survey period (S4 Table), when recruit abundance varied substantially across tiles at the lagoonal site. At the lagoonal site, mean abundance peaked in October–December 2018 (4080 ± 431 recruits m$^{-2}$) and was lowest in June–August 2019 (382 ± 164 recruits m$^{-2}$). At the seaward site, mean abundance was higher in October–December 2018 (244 ± 58 recruits m$^{-2}$) and December 2018–February 2019 (342 ± 98 recruits m$^{-2}$) than the remaining time periods (31–60 recruits m$^{-2}$).

Pocilloporidae comprised 74% of all recruits at the lagoonal site, which varied little throughout the study period (range: 67–93%). Acroporidae comprised 18% of the overall counts at the lagoonal site (21–25% in August 2018–February 2019; 1–6% in February–August

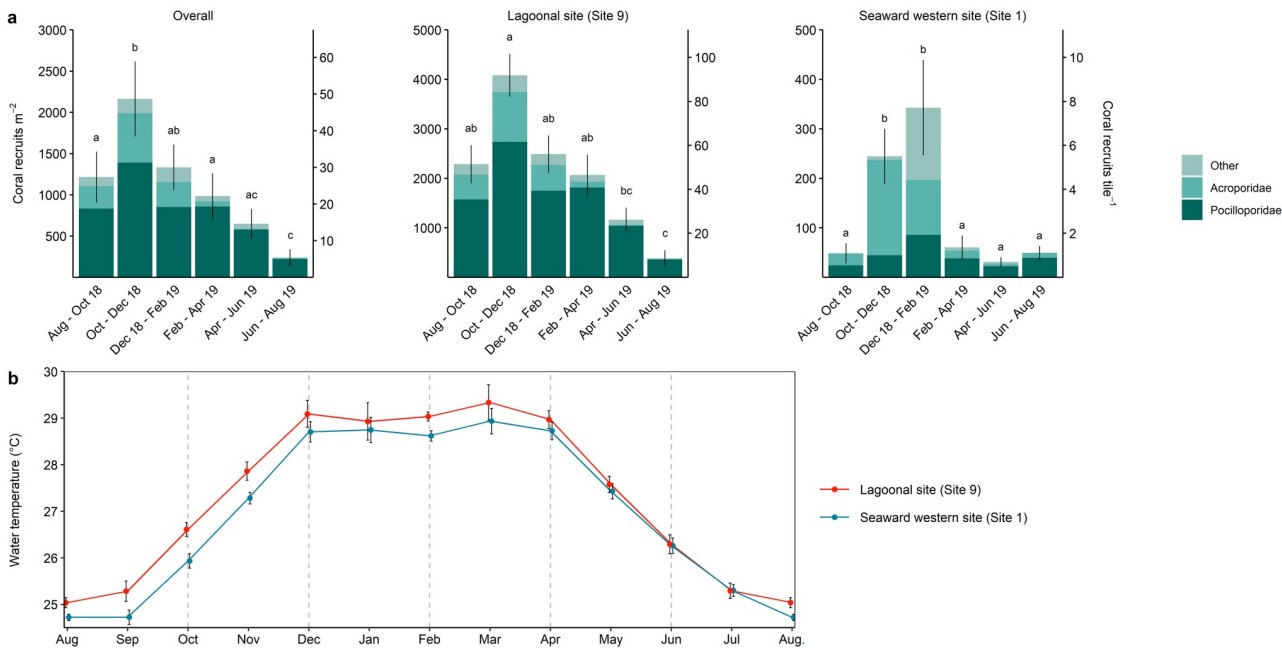

**Fig 4. Results of coral recruit study.** (A) Coral recruit abundances overall and at each site. Values represent mean abundance for each taxon and in total, SE is displayed for total mean abundance only. Secondary y-axis indicates number of coral recruits per tile. Note the different y-axis scaling. Letters indicate significant differences ($p < 0.05$) between study periods (see S4 Table). (B) Mean (± SE) monthly water temperature during December 2013–Dec 2019 at the seaward western site and during February 2015–November 2018 at the lagoonal site where technical issues prevented the inclusion of data until December 2019. Horizontal dashed lines indicate month of settlement tile change-over.

2019). At the seaward site, Pocilloporidae and Acroporidae comprised 32% and 45% of overall counts, respectively, but while the proportion of Pocilloporidae varied little throughout the study periods, Acroporidae abundance peaked in October–December (79% of overall count in this period) and was higher in December 2018–February 2019 (32%) than the remaining periods (14–26%). The contribution of 'other' recruits was lower at both sites (8% and 23% of overall counts at the lagoonal and seaward site, respectively), peaking at the seaward site in December 2018–February 2019 (43% of total counts).

The temporal variation in coral recruit abundances coincided with seasonal changes in water temperatures (Fig 4b). At both sites, abundance of Acroporidae recruits was highest when water temperature increased during October–December. At the seaward site, abundances of 'other' recruits peaked when water temperatures remained around the maximum monthly mean for three months (December 2018–February 2019).

## Discussion

Governed by a variety of interacting physical and biological processes, abundance and composition of coral juveniles often show high spatio-temporal variability, even in the absence of major disturbances [48, 49]. While recognising that our 2015 data is limited in its suitability to account for the natural temporal variability in coral juvenile abundance (i.e., low replication at deep sites and no surveys in the lagoon due to limited resources), our results nonetheless indicate a substantial decrease in the abundance of coral juveniles following the 2015/2016 bleaching event at Aldabra (48% reduction). This percentage loss is remarkably similar to reductions in overall hard coral cover (50% [33, 34]), and matches observations from the Inner Seychelles where the reduction of coral juveniles and coral cover over the same time period was also similar (i.e., *ca.* 70% reduction of both [50, 51]). While temperature stress and bleaching can directly lead to coral mortality, it also affects the reproductive capacity of corals [52–54]. Reduced fecundity, as a result of temperature stress and/or bleaching, can affect corals for several years after the disturbance, consequently leading to lower coral recruitment in subsequent years [52]. This, in addition to bleaching-induced mortality of recruits, juvenile and adult corals, may have contributed to the reduced density of coral juveniles following the bleaching event at Aldabra.

The reductions in abundance of coral juveniles during 2015–2016 showed high spatial variability at Aldabra; however, absolute abundances in 2016 (six months after the bleaching event) were similar across locations. This may suggest that spatial variation in juvenile coral losses reflects differences in pre-bleaching abundance and composition of juvenile corals, and potentially variable effects of temperature stress on the reproductive capacity of adult corals that survived the bleaching event. Taxonomic variations in bleaching susceptibility and mortality have been widely reported [55–58] and several studies have shown that temperature stress affects coral fecundity differently among taxa and coral colony sizes [e.g., 51, 52, 57]. In the present study, Acroporidae, Agariciidae, and Merulinidae juveniles exhibited the greatest decreases in abundance after the bleaching event, while Poritidae juvenile abundance slightly increased. These changes partially overlap with percentage cover changes of Aldabra's most abundant coral taxa: Acroporid (*Acropora*; encrusting *Montipora*) and Merulinidae cover decreased by 85% and 57%, respectively, following bleaching, while the reduction of massive *Porites* cover was lower (13% [33]; but see [34], showing a distinct reduction of branching *Porites*). These trends somewhat deviate from those in the Inner Seychelles. The loss of branching corals, which predominate the reefs there, largely drove the 70% reduction in coral cover observed, with *Acropora*, *Pocillopora* and *Porites* being particularly affected by bleaching and mortality [50, 51]. Changes in percentage cover of Pocilloporidae, Agariciidae, and *Leptastrea*

(commonly recorded as juveniles at Aldabra) were not assessed previously at Aldabra, due to their low contribution to overall benthic cover (S1 Fig). This may indicate limited correlation between juvenile and overall coral community composition at Aldabra, similar to observations made in the Netherlands Antilles [59] and on the Great Barrier Reef [48], but contrary to e.g., reefs in French Polynesia (Moorea [49]) and New Caledonia [60]. Strong overlap in juvenile and adult coral assemblages may indicate that adult assemblages are partially controlled by the history of recruitment patterns or *vice versa* [49, 60], although authors note that this is taxon dependent, likely resulting from variable life histories. However, such relationships may ultimately be disrupted by a large-scale disturbance such as coral bleaching [61], whereby the early post-bleaching coral juvenile composition observed at Aldabra may represent an early successional stage which is not (yet) reflected in the overall coral community composition.

As hard coral recovery can speed up exponentially with ongoing coral recruitment [6], the abundance of coral juveniles is widely recognised as an important indicator of reef recovery. Our research provides valuable baseline information on coral juvenile patterns at Aldabra. Although future research is necessary to elucidate more robust recovery patterns, our study provides valuable insights into early recovery trajectories following the bleaching event. Post-bleaching coral juvenile abundances at most of Aldabra's locations exceeded 6.2 juveniles m$^{-2}$; an amount identified as critical threshold on reefs within the Inner Islands of the Seychelles, above which reef recovery is more likely to occur [2]. In addition, overall abundance of coral juveniles tripled at Aldabra during 2016–2019 and absolute abundances in 2019 were high (16.6–29.9 coral juveniles m$^{-2}$), in comparison to both Aldabra's abundances recorded in 2015 (i.e., pre-bleaching; 8.3–20.6 coral juveniles m$^{-2}$), and other protected and unprotected locations: 83% of the sites (i.e. 75 of 90 sites from eight studies) in S5 Table had abundances of $\leq$16 coral juveniles m$^{-2}$, whilst only 3% had abundances $\geq$ 30 coral juveniles m$^{-2}$ (S2 Fig). This increase has probably been facilitated by rapid recovery of the reproductive capacity of Aldabra's coral stock (and connected reefs [62, 63]), coupled with the availability of suitable settlement substrate (such as CCA) and sufficient herbivory [34].

Post-disturbance reef recovery is often driven by branching hard corals of the families Acroporidae (e.g., *Acropora* in the Inner Seychelles [51]; Scott Reef, Australia [6]) and Pocilloporidae (e.g., *Pocillopora* in Kenya [57]; Moorea, French Polynesia [60]) displaying life-history traits characteristic of 'competitive' and 'weedy' species [64], such as rapid growth [51, 60]. Previous work at Aldabra identified branching *Porites* (lagoon) and encrusting *Montipora* (seaward reefs) as key drivers of early hard coral cover increase during 2016–2019, likely driven by re-growth of remnant colonies [34]. In contrast, juvenile coral communities at Aldabra were not dominated by individual coral taxa in any of the years studied (unlike e.g., reefs in the Maldives where *Acropora* juveniles dominated [65], or the Inner Seychelles with *Acropora*, *Porites*, and *Pocillopora* dominating [20]), and juvenile abundances of all taxa (except Merulinidae) reached, or even exceeded, pre-bleaching levels at all locations by 2019. The increase of Pocilloporidae and *Leptastrea* juveniles at Aldabra, however, is conspicuous, as their percent cover in 2014–2019 was low (S1 Fig). Pocilloporidae (i.e., the genera *Pocillopora*, *Stylophora*, *Seriatopora*), some *Leptastrea* species and also branching members of Poritidae (*Porites* species) follow competitive or weedy life-history strategies [64]. Weedy corals are generally characterised by small colony sizes, fast growth, and low resistance to perturbations, and can opportunistically colonise recently disturbed habitats [64]. They often reproduce by brooding larvae, with some having the ability to produce larvae asexually by parthenogenesis [66, 67], allowing successful reproduction even if the population of corals at reproductive age is small [64]. The increased abundance of Pocilloporidae, *Lepastrea*, and Poritidae juveniles observed at Aldabra between 2016 and 2019 thus likely arose from weedy species of these taxa.

Although coral juvenile abundances increased across all locations and water depths, increases at the lagoonal and western sites occurred earlier than at the more exposed eastern site. Spatial variation was also recorded for the recovery of hard coral cover [34], with significant increases only at the shallow reefs and by a substantially lower magnitude at the eastern sites compared with the other locations (Table 1). This slow recovery at the eastern sites may have been linked to high *Halimeda* cover (45–61% in 2016–2019 [34]), which was already noted in the 1970s and attributed to the high hydrodynamic energy the eastern reefs are exposed to [68, 69]. Although the presence of *Halimeda* can increase post-settlement mortality of coral recruits [70], thereby suppressing juvenile abundances, the high abundance of coral juveniles at the eastern sites in 2019 (despite high *Halimeda* cover) does not entirely support the assumption that *Halimeda* is the sole driver for low coral juvenile abundance there up until 2019. It is possible that Aldabra's eastern sites are typically more exposed to oceanic waters devoid of coral larvae than the other locations and that a change in environmental conditions may have favoured coral recruitment and growth of juvenile corals at the eastern sites by 2019.

Spatial variation was also pronounced for the abundance of coral recruits on settlement tiles, which is in line with several other studies (e.g., [71–73]). As the recruit abundance data in the present study only spans a 12-month period, further research is needed to confirm the observed patterns, which represent baseline information for Aldabra. Recruit abundance at Aldabra's lagoonal site was seven- to 47-fold higher than at the seaward site, where exposure to wind, swell, and currents may reduce larval settlement rates and/or increase post-settlement mortality (e.g., [72, 74]). Indeed, this western seaward site is close to Aldabra's West Channels where water current velocities reach 1.5 m s$^{-1}$ [75], and is possibly influenced by eddies forming on the north-western side of the atoll [76]. Coral gamete and larval dispersal may be enhanced on the seaward reefs compared to the lagoon, where complete water tidal exchange amounts to 18 and 53 h at spring and neap tides, respectively [77]. Most of the recruits counted in the lagoon, particularly Pocilloporidae, may therefore be of local (i.e., lagoonal) origin. High abundance of Pocilloporidae recruits despite low Pocilloporidae cover (S2 Fig) may result from parthenogenesis [64, 66, 67], but could also originate from non-surveyed areas in the lagoon with higher density of Pocilloporidae colonies.

Overall recruit abundances in Aldabra's lagoon (2080 ± 193 recruits m$^{-2}$) were substantially higher than those reported at other lagoonal reefs (i.e., annual mean of 101–908 recruits m$^{-2}$ post-1998 bleaching in Kenya [73]; means of 225–780 recruits m$^{-2}$ pre-2016 bleaching in Fiji [37]). However, the abundances at Aldabra's seaward site (141 ± 26 recruits m$^{-2}$) were low compared to those recorded at seaward reefs in the Inner Seychelles (means of 355–832 recruits m$^{-2}$ pre-2016 bleaching [20]), Vamizi Island, Mozambique (annual mean of 1135 recruits m$^{-2}$ pre-2016 bleaching [78]), the Spermonde Archipelago (annual mean of 286–686 recruits m$^{-2}$ [79]) and French Polynesia (mean of 569 recruits m$^{-2}$ [80]), and closer to values reported from the Mascarene Islands, Réunion and Rodrigues (maximum 40–150 recruits m$^{-2}$ post-2016 bleaching [81]). Comparisons, however, should be treated with caution due to different methods used [82, 83] and high spatial and temporal variability of recruitment [71, 84].

Spatial discrepancies between the lagoonal and seaward sites of Aldabra are also reflected in the sites' coral juvenile abundances and percentage coral cover, both of which were considerably higher at the lagoonal than the seaward site (Table 2). Inferences drawn from the assessment of single sites include substantial speculation. However, taken together, the patterns in coral recruit abundance, juvenile abundance, and coral coverage suggest that, compared to the seaward reefs, Aldabra's lagoonal reefs appeared less susceptible to the last coral bleaching event [33, 34], and seem to recover rapidly by fast regrowth of remnant corals [34] and high rates of coral recruitment (Table 1). Whether or not the lagoonal reefs may benefit Aldabra's

entire reef system is currently not known, and research on connectivity patterns among Aldabra's reef sites is needed.

The results of the coral recruit study provide important baseline data for further research. In common with tropical and sub-tropical reefs across the Indo-Pacific [20, 37, 72, 81, 82, 85, 86], Pocilloporidae recruits dominated at Aldabra and likely resulted predominantly from brooding corals [87, 88], which release planulae all year round [89]. Contrastingly, broadcast spawners (e.g., many members of the Acroporidae family) release gametes typically when environmental conditions are optimal for fertilisation, larval survival, and settlement [90], resulting in seasonal variation [48]. Coral recruit abundance at Aldabra (most notably of Acroporidae) was highest during October–December, suggesting that broadcast spawning peaks at the beginning of the north-west monsoon, similar to observations from Kenya [91]. Although multi-annual recruitment studies and/or direct observations of coral reproduction are necessary to confirm the spawning patterns for a given location (e.g., [92]), the October–December time frame coincides with the period of most rapid seasonal increase in water temperature at Aldabra, which likely serves as proximate cue for broadcast spawning [73, 92–94]. This may also suggest that broadcast spawning at other reefs in the immediate region occurs during a similar time frame; information that is crucial for an improved understanding of regional reef connectivity [63]. Recent modelling studies suggest that Aldabra receives and provides coral larvae from/to other Outer Islands of the Seychelles [62] and that it is a stepping stone for larvae and gene dispersal in the Western Indian Ocean [63], highlighting Aldabra's regional importance.

Our results are the first published record on coral recruitment in the Seychelles Outer Islands. The rapid increase of juvenile corals of Aldabra's reefs after bleaching, particularly in the lagoon, suggests that similar trajectories may occur at other remote reefs in the region and that managing local anthropogenic pressures, such as overfishing, coastal development and pollution, is critical to promote reef recovery following coral bleaching events. Continued monitoring of the recovery of reefs in this region as well as an improved understanding of their connectivity will therefore be invaluable for the conservation of Aldabra's reefs and beyond. Nevertheless, although reef recovery at Aldabra appears to be fast, those corals that increased in abundance are also some of the most susceptible to bleaching [8, 95, 96]. With an increasing frequency and severity of coral bleaching events [97], reefs in the early stages of recovery may thus be highly susceptible to the effects of temperature stress [61].

## Supporting information

**S1 Fig. Percentage cover of different hard coral taxa and overall between 2014 and 2019 across Aldabra's locations.** Plots show median and interquartile range with outliers displayed as circles and means displayed as red dots. Note that no data is available from 2015 and the pre–bleaching reference is the 2014 data.
(TIF)

**S2 Fig. Comparison of Aldabra's coral juvenile abundance in 2019 with pre-bleaching values at Aldabra and reefs worldwide.** The number (percentage) of sites (blue marked values in S5 Table) with coral juvenile abundances that fall within the same range as Aldabra's reefs in 2019 (16.1–29.9 coral juveniles m$^{-2}$) or below/above.
(DOCX)

**S1 Table. Coral juvenile survey replicates.** Number of transects and quadrats completed at each location and site during each survey year. Lower number of replicates obtained in 2015 due to limited resources.
(DOCX)

**S2 Table. Coral recruit survey replicates.** Number of coral settlement tiles retrieved per site and survey period. Replicates at the seaward site vary as some tiles were lost due to rough weather.
(DOCX)

**S3 Table. Change in coral juvenile abundances.** Effect of year (2015, 2016, 2018, 2019), location (lagoon, western seaward, eastern seaward) and depth (2 m, 5 m, 15 m) on coral juvenile abundances at Aldabra. Chi-square-value ($\chi^2$), degrees of freedom (dF) and $p$-value obtained from GLMM model comparisons with ANOVA (type II). Significance level: *** $p < 0.001$; ** $p < 0.01$; ** $p < 0.05$; ns = not significant: $p > 0.05$.
(DOCX)

**S4 Table. Difference in coral recruit abundances.** Effect of location (seaward western site: Site 1, lagoonal site: Site 9) and time period (Aug–Oct 2018, Oct–Dec 2018, Dec 2018–Feb 2019, Feb–Apr 2019, Apr–Jun 2019, Jun–Aug 2019) on coral recruit abundances on settlement tiles. Chi-square-value ($\chi^2$), degrees of freedom (dF) and $p$-value obtained from GLMM model comparisons with ANOVA (type II). Results of the pairwise tests were derived from post-hoc analysis based on least square means with Bonferroni adjustment. Significance level: *** $p < 0.001$; ** $p < 0.01$.
(DOCX)

**S5 Table. Coral juvenile abundances (per m$^2$) reported from 106 reefs at 11 locations worldwide.** A: Studies reporting coral juvenile abundances before and/or after a bleaching event. B: Studies reporting coral juvenile abundance without referring to a bleaching event. Values in blue are those included in S2 Fig (comparison of Aldabra's coral juvenile abundance in 2019 to pre-bleaching values at Aldabra and reefs worldwide). 'Nr. years studied' refers to number of years data was collected within each study.
(DOCX)

**S1 Data.**
(XLSX)

**S2 Data.**
(XLSX)

**S3 Data.**
(CSV)

**S4 Data.**
(CSV)

## Acknowledgments

We thank all former and current staff of the Seychelles Islands Foundation (SIF) for their work and support. Specifically, we thank Jude Brice, April J Burt, Karen Chong-Seng, Marvin Roseline, Joel Bonne, Shane Brice, Ella Nancy, Leeroy Estrale, Ronny Marie, Jake Letori, Lorraine Cook, Jilani Suleman, Germano Soru, Diane Ernesta and Luke A'Bear for data collection and support in the field. A special thanks also goes to Jessica Constance for the analysis of settlement tiles. We further thank the Global Environment Facility for supporting the development of the Aldabra Reef Monitoring programme. Thanks also go to Philip Haupt, Anthony Bernard and Nick Riddin from the South African Institute for Aquatic Biodiversity and SIF staff for establishing the permanent transects.

## Author Contributions

**Conceptualization:** Anna Koester, Amanda K. Ford, Nancy Bunbury, Christian Wild.

**Data curation:** Anna Koester, Valentina Migani, Cheryl Sanchez.

**Formal analysis:** Anna Koester, Valentina Migani.

**Investigation:** Anna Koester, Cheryl Sanchez.

**Methodology:** Anna Koester, Amanda K. Ford, Valentina Migani, Cheryl Sanchez.

**Project administration:** Anna Koester, Nancy Bunbury, Cheryl Sanchez.

**Supervision:** Nancy Bunbury, Christian Wild.

**Validation:** Anna Koester, Amanda K. Ford, Sebastian C. A. Ferse, Christian Wild.

**Visualization:** Anna Koester.

**Writing – original draft:** Anna Koester.

**Writing – review & editing:** Anna Koester, Amanda K. Ford, Sebastian C. A. Ferse, Valentina Migani, Nancy Bunbury, Cheryl Sanchez, Christian Wild.

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
