## [Decision Letter · Decision Letter 0]

28 Jul 2021

PONE-D-21-19918

First insights into coral recruitment and juvenile abundances at remote Aldabra Atoll, Seychelles

PLOS ONE

Dear Dr. Koester,

Thank you for submitting your manuscript to PLOS ONE. After careful consideration, we feel that it has merit but does not fully meet PLOS ONE’s publication criteria as it currently stands. Therefore, we invite you to submit a revised version of the manuscript that addresses the points raised during the review process.

Dear Authors,

As you can see, both the reviewers have recommended minor revisions, especially reviewer 2 has provided detailed review with suggestions and comments. This will help make the manuscript better and suitable for PLoS one

I look forward to the revised version of the manuscript

We look forward to receiving your revised manuscript.

Kind regards,

Shashank Keshavmurthy, PhD

Academic Editor

PLOS ONE

Journal Requirements:

2. We note that Figure 1 in your submission contain map images which may be copyrighted. All PLOS content is published under the Creative Commons Attribution License (CC BY 4.0), which means that the manuscript, images, and Supporting Information files will be freely available online, and any third party is permitted to access, download, copy, distribute, and use these materials in any way, even commercially, with proper attribution. For these reasons, we cannot publish previously copyrighted maps or satellite images created using proprietary data, such as Google software (Google Maps, Street View, and Earth). For more information, see our copyright guidelines: http://journals.plos.org/plosone/s/licenses-and-copyright.

Reviewers' comments:

Reviewer's Responses to Questions

**Comments to the Author**

1. Is the manuscript technically sound, and do the data support the conclusions?

Reviewer #1: Yes

Reviewer #2: Yes

2. Has the statistical analysis been performed appropriately and rigorously? 

Reviewer #1: Yes

Reviewer #2: Yes

3. Have the authors made all data underlying the findings in their manuscript fully available?

Reviewer #1: Yes

Reviewer #2: Yes

4. Is the manuscript presented in an intelligible fashion and written in standard English?

Reviewer #1: Yes

Reviewer #2: Yes

5. Review Comments to the Author

Reviewer #1: The manuscript described the changes in coral juvenile abundance before/after coral bleaching event, and compared between 12 different sites in atoll. Also, authors compared the coral recruitment abundance between lagoon and seaward sites by settlement tile. All of the experiments were conducted at remote Aldabra Atoll where no human disturbance occurs. This manuscript contributes to the more understanding of early life stage dynamics of coral. The manuscript is well written and the results presented here are worthy of publication in PLOS One. I only have few comments for this manuscript.

In Discussion, I feel the authors did not provide satisfactory explanation about the decrease of coral juveniles after bleaching event. I can understand the results showed similar patterns between coral coverage and juvenile abundance, however, many factors could cause the decrease of coral juveniles, for example, bleaching also influence the coral reproduction for several years, and the recovery patterns are similar as the results in this study (see Levitan et al. 2014 and Johnston et al. 2020).

Reviewer #2: Review of Koester, Ford, Ferse, Migani, Bunbury, Sanchez and Wild “First insights into coral recruitment and juvenile abundances at remote Aldabra Atoll, Seychelles”

The study of Koester, Ford, Ferse, Migani, Bunbury, Sanchez and Wild assessed coral recruitment in the Aldabra Atoll of the Outer Seychelles by looking at two vital life cycle stages, coral recruits and coral juveniles.

Koester et al. give interesting and important insights into the impact of the 2016 mass coral bleaching event on Aldabra Atoll and the recruitment dynamics after it. They also assess more limited datasets before the event. The reported coral recovery of the remote atoll is remarkably rapid and indicates the potential of corals to respond to bleaching events with reduced presence of human stressors. This is an important insight to see published. Koester et al.’s interpretations of the findings are on point; I especially enjoyed their interpretations of their results and the corresponding hypotheses as to why they could be seen. I have minor suggestions for improvements.

The manuscript is carefully prepared and well-written. The most important of my improvement suggestions are regarding the structure of introduction and discussion. Some sentences are very long as well, I have marked the ones that definitely need breaking up. Please find specific comments below.

Specific comments

Introduction:

General – Sometimes, ‘recruit’ and ‘recruitment’ get used interchangeably in the text. For instance, in line 200: “As the coral recruitment dataset may have an excess of zeros”. Should this not be the ‘coral recruit dataset’?

To my knowledge, ‘coral recruitment’ is the replenishment of the local adult population by new individuals from within or outside an existing population (Hughes et al. 2010). I always understood this as coral juveniles being included in the recruitment process. Therefore, I would be careful when treating the words interchangeably.

Hughes TP, Graham NAJ, Jackson JBC, Mumby PJ, Steneck RS (2010) Rising to the challenge of sustaining coral reef resilience. Trends in Ecology & Evolution 25:633–642

General – I do not think you need to introduce recruitment dynamic topics like rubble and CCA in the introduction, since neither are picked up in the rest of the manuscript. I would delete the phrases regarding both CCA and rubble.

Methods:

L136 – Is the sentence regarding approval necessary in a manuscript? I have not had the need to include it yet.

L145 – I think you might need a reference for that fragmented colonies statement.

L148-153 – Very long sentence, needs breaking up.

L198 – The comma after “i.e.” can be removed.

L200-204 – Very long sentence, twice with “which” in it too, needs breaking up.

L204 – I would clarify that you did not need to run a zero-inflated model for the juveniles in the end.

L214 – I have found model validation with glmmTMB to be difficult because I was unable to extract Pearson residuals which are needed for dispersion testing (this was a couple of years back though, might be updated now). How did you get around that issue?

L225 – That table is super useful but unfortunately not picked up again anywhere. That makes it feel a bit redundant. What was the aim in creating it? I would mention it somewhere, it might save a lot of researchers some time if it’s more in the spotlight.

Results:

General – the selection of which sites are presented throughout the text seems arbitrary at times and does not really follow a, for me, recognisable pattern.

Discussion:

General – I am not a fan of the first paragraph of the discussion. The rest of the discussion is great, especially the conclusion and the hypotheses you formulate for your findings! Just the first paragraph is lacking the clarity and punchiness of the rest.

L331-340 – This text should really be in the introduction. If the first discussion paragraph should include any introductory content it should only be one sentence at most.

Then you can get right into the most important result of your study, then move onto the next most important finding, and the next, and so on. Each finding should really only have one or two sentences. I would then structure the rest of the discussion in the same order of your first paragraph, each ensuing paragraph explains one finding, like you have already. I would just rearrange the paragraphs.

L355 – Oxford comma missing (the comma before the ‘and’). Most of your text uses commas before ‘and’ in an enumeration, which is fine, just needs to be consistent throughout the text.

L384 – It would help to mention the location of the Graham et al. study here; I think it would help the reader contextualising.

L409 – You can drop the apostrophes around ‘weedy’ here, you have introduced it in the text before already.

L415 – Oxford comma missing.

L465 – Oxford comma missing

L463-469 – Really long sentence with ‘… reefs, reefs …’ in the middle, makes it tough to read. I would break it up.

If anything is unclear, please do not hesitate to contact me.

Best,

Jan-Claas Dajka

6. PLOS authors have the option to publish the peer review history of their article (what does this mean?). If published, this will include your full peer review and any attached files.

Reviewer #1: No

Reviewer #2: **Yes: **Jan-Claas Dajka

---

## [Author Response · Author response to Decision Letter 0]

4 Sep 2021

Point-to-point responses to the reviewers are provided in a separate file labelled "Response to Reviewers".

---

## [Decision Letter · Decision Letter 1]

12 Nov 2021

First insights into coral recruit and juvenile abundances at remote Aldabra Atoll, Seychelles

PONE-D-21-19918R1

Dear Dr. Koester,

We’re pleased to inform you that your manuscript has been judged scientifically suitable for publication and will be formally accepted for publication once it meets all outstanding technical requirements.

Kind regards,

Shashank Keshavmurthy, PhD

Academic Editor

PLOS ONE

Additional Editor Comments (optional):

Reviewers' comments:

Reviewer's Responses to Questions

**Comments to the Author**

1. If the authors have adequately addressed your comments raised in a previous round of review and you feel that this manuscript is now acceptable for publication, you may indicate that here to bypass the “Comments to the Author” section, enter your conflict of interest statement in the “Confidential to Editor” section, and submit your "Accept" recommendation.

Reviewer #1: (No Response)

Reviewer #2: All comments have been addressed

2. Is the manuscript technically sound, and do the data support the conclusions?

Reviewer #1: Yes

Reviewer #2: Yes

3. Has the statistical analysis been performed appropriately and rigorously? 

Reviewer #1: Yes

Reviewer #2: Yes

4. Have the authors made all data underlying the findings in their manuscript fully available?

Reviewer #1: Yes

Reviewer #2: Yes

5. Is the manuscript presented in an intelligible fashion and written in standard English?

Reviewer #1: Yes

Reviewer #2: Yes

6. Review Comments to the Author

Reviewer #1: (No Response)

Reviewer #2: Koester and colleagues carefully addressed all comments and thoroughly answered every question I had regarding their study. I fully support the publication of their work.

Thank you very much for the pleasent review process.

Best,

Jan-Claas Dajka

7. PLOS authors have the option to publish the peer review history of their article (what does this mean?). If published, this will include your full peer review and any attached files.

Reviewer #1: No

Reviewer #2: **Yes: **Jan-Claas Dajka

---

## [Editor Report · Acceptance letter]

18 Nov 2021

PONE-D-21-19918R1 

First insights into coral recruit and juvenile abundances at remote Aldabra Atoll, Seychelles 

Dear Dr. Koester:

I'm pleased to inform you that your manuscript has been deemed suitable for publication in PLOS ONE. Congratulations! Your manuscript is now with our production department. 

Kind regards, 

on behalf of

Dr. Shashank Keshavmurthy 

Academic Editor

PLOS ONE